# Analysis of RNA Polyadenylation in Healthy and Osteoarthritic Human Articular Cartilage

**DOI:** 10.3390/ijms24076611

**Published:** 2023-04-01

**Authors:** Phaedra Winstanley-Zarach, Gregor Rot, Shweta Kuba, Aibek Smagul, Mandy J. Peffers, Simon R. Tew

**Affiliations:** 1Centre for Integrated Research into Musculoskeletal Ageing (CIMA), Department of Musculoskeletal and Ageing Science, Institute of Life Course and Medical Sciences, University of Liverpool, Liverpool L7 8TX, UK; 2Institute of Molecular Life Sciences, Winterthurerstrasse 190, 8057 Zurich, Switzerland; 3Swiss Institute of Bioinformatics, Amphipôle, Quartier UNIL-Sorge, 1015 Lausanne, Switzerland; 4School of Health and Life Sciences, National Horizons Centre, Teesside University, Darlington DL1 1HG, UK

**Keywords:** polyadenylation, osteoarthritis, cartilage, mRNA, RNA-Seq

## Abstract

Polyadenylation (polyA) defines the 3′ boundary of a transcript’s genetic information. Its position can vary and alternative polyadenylation (APA) transcripts can exist for a gene. This causes variance in 3′ regulatory domains and can affect coding sequence if intronic events occur. The distribution of polyA sites on articular chondrocyte transcripts has not been studied so we aimed to define their transcriptome-wide location in age-matched healthy and osteoarthritic knee articular cartilage. Total RNA was isolated from frozen tissue samples and analysed using the QuantSeq-Reverse 3′ RNA sequencing approach, where each read runs 3′ to 5′ from within the polyA tail into the transcript and contains a distinct polyA site. Differential expression of transcripts was significant altered between healthy and osteoarthritic samples with enrichment for functionalities that were strongly associated with joint pathology. Subsequent examination of polyA site data allowed us to define the extent of site usage across all the samples. When comparing healthy and osteoarthritic samples, we found that differential use of polyadenylation sites was modest. However, in the genes affected, there was potential for the APA to have functional relevance. We have characterised the polyadenylation landscape of human knee articular chondrocytes and conclude that osteoarthritis does not elicit a widespread change in their polyadenylation site usage. This finding differentiates knee osteoarthritis from pathologies such as cancer where APA is more commonly observed.

## 1. Introduction

Osteoarthritis is a multifactorial disease in which initiation and progression is driven by a combination of genetic, environmental and mechanical factors. The functionality of the chondrocytes within the articular cartilage of the joint is significantly altered by the late stages of the disease, and this is associated with altered gene-expression patterns that have been examined in a range of earlier transcriptomic studies [1].

The processes regulating gene-expression changes during osteoarthritis are a source of ongoing study. Genetic-association studies have identified numerous polymorphisms that are linked to osteoarthritis incidence [2], which can affect the processes that control transcriptional regulation [3]. Mechanisms of transcriptional regulation of known effectors of articular cartilage degradation, such as extracellular proteases, have also been determined [4]. Transcriptomic analysis of osteoarthritic cartilage in a variety of species has demonstrated that widespread changes in gene expression are induced by the disease [5].

A growing number of studies have implicated post-transcriptional gene-regulation in the disruption of transcript abundance during osteoarthritis. Mouse genetic models have identified significant disruption to skeletal development when regulators of mRNA decay or translation such as microRNAs and RNA binding proteins are disrupted [6,7,8], indicating the importance of these factors in chondrocyte function. Our previous studies have demonstrated that mRNA decay rates are affected in osteoarthritic human chondrocytes, with an increased number of short-lived mRNAs evident in osteoarthritic cells [9].

One aspect of post-transcriptional gene-regulation that has received little attention in cartilage biology is the alternative polyadenylation (APA) of transcripts. It is well established that a high proportion of human genes display evidence of APA, leading to functional and structural heterogeneity in the transcript pool [10]. By using different polyadenylation sites, transcripts from the same gene can be produced which have altered 3′ untranslated region sequence lengths. This can result in the inclusion or exclusion of protein and small RNA interacting regions, generating pools of transcripts that code for the same protein but have different regulatory responses [11]. Functionally, altered 3′UTR length induced by APA influences the decay, localisation and translation of mRNAs. Furthermore, the use of polyadenylation sites upstream of the 3′UTR, often contained within intronic sequences, can lead to transcripts where the protein-coding region is altered resulting in the production of truncated proteins with altered functionality [12,13].

Evidence now suggests that modulation of the transcriptome by APA can be dynamic, is driven by differentiation state and can be tissue-specific [14]. Furthermore, links between APA and pathologies are beginning to emerge. Global shortening of 3′UTRs because of APA has been demonstrated to be a feature of cancer pathology [15,16] whilst impaired APA has been implicated in the development of alcohol-induced fatty liver disease [17]. To date however, there has been no study into how chondrocytes utilise this method of transcript mRNA regulation or whether it is affected by degenerative diseases such as osteoarthritis.

In this study we used the QuantSeq-Reverse 3′ end sequencing approach, which produces reads that run 3′ to 5′ from the polyA tail into the 3′ transcript sequence [18], to examine RNA purified from healthy or osteoarthritic human knee articular chondrocytes. By using age-matched human samples these data allow complementation of existing studies by analysis of how transcript levels change in the disease. Furthermore, these data also allow us to define how chondrocytes polyadenylate their transcripts and how this is affected by disease. We determined that chondrocytes, like many other cell types, make widespread use of APA to generate diversity in their transcript pool. We also found that disease-associated changes in APA are observed at a relatively low level but that there is potential to produce significant alterations to both gene function and regulatory kinetics where it occurs.

## 2. Results

### 2.1. Analysis of Transcript Abundance in Control and Osteoarthritis Articular Cartilage Using QuantSeq Reverse

We used the QuantSeq-Reverse approach to create 3′ RNA sequence libraries that provided transcriptome-wide read coverage across human chondrocyte polyadenylation sites. Total RNA isolated directly from snap-frozen tissue was used for this and mean sample age for healthy tissues was 75 years with standard deviation (SD) of ± 6 (n = 6) and for osteoarthritic tissues mean age was 70 years with SD± 5 (n = 6). Analysis with an unpaired *t*-test indicated no statistical difference between the ages of the groups (*p* = 0.16). Sequencing resulted in median sample reads of 4.4 × 10^7^ with the highest number of reads in a sample being 6.0 × 10^7^ and the lowest 3.7 × 10^7^ reads. We began our analysis by examining differential expression of steady-state transcript levels between healthy and osteoarthritic samples (Figure 1). 

Principal component analysis of the data demonstrated that the first principal component (PC) was responsible for 40% of the experimental variance and the second principal component, 13% of the variance. There was a clear separation between the healthy and osteoarthritic samples across PC1 (Figure 1A). Analysis of differential expression revealed that osteoarthritic tissue had 561 transcripts that were differentially upregulated and 630 transcripts that were differentially downregulated (where log2fc > 1 and adjusted *p-*value was < 0.05) (Figure 1B and Appendix A). There was overlap with OA-regulated genes observed in other studies. For instance, when comparing genes regulated in a study of non-OA vs. OA human tissue [5], we found that 423 (31%) of the genes we identified as being significantly regulated were also regulated in the other dataset with strong correlation in the direction of the fold changes (Figure 1C). Functional annotation enrichment analysis revealed a strong association of regulated genes with osteoarthritis-related characteristics such as extracellular structure/matrix organisation, and skeletal development in addition to associations with cell motility and migration (Figure 1D). Examination of differentially regulated genes using RcisTarget allowed us to identify enriched transcriptional regulatory sequences associated with OA (Figure 1E). This identified that genes with promoter sequences containing SMAD3 and IKZF1 binding motifs were over-represented in genes that were regulated during osteoarthritis. Regulated genes also consisted of a significant number of long non-coding RNAs that were differentially regulated in osteoarthritis (Figure 1F). We noted that very few small regulatory RNAs such as microRNAs (miRNAs) or small nucleolar RNAs (snoRNAs) were represented in the data set. When examining data from across both control and osteoarthritic samples, only 4 miRNAs (*miR125b-1*, *miR320d-1*, *miR3671*, *miR7851*) and 11 snoRNAs (*SNORA11F*, *SNORA12*, *SNORA59B*, *SNORA72*, *SNORA79B*, *SNORD104*, *SNORD13*, *SNORD13E*, *SNORD38C*, *SNORD38-1*, *SNORD38-2*) were detected across more than 50% of the samples analysed. No significant effect of osteoarthritis was observed on the levels of the small RNAs detected with *SNORD13* which exhibited a log_2_ fold decrease of 2.0 in osteoarthritic samples with an adjusted *p*-value of 0.093, being the only one to exhibit a trend toward regulation by disease.

### 2.2. Human Articular Chondrocytes Make Widespread Use of Alternative Polyadenylation to Modify Their Transcripts

Making use of the Quantseq-Reverse data property of sequencing 3′–5′ from the polyA tail, we were able to use the dataset to identify chondrocyte polyadenylation site usage. Aligned reads were analysed using expressRNA to identify polyadenylation site usage in each sample. A full list of polyadenylation sites identified by this analysis can be found in Appendix A. To understand general chondrocyte polyadenylation site usage, the healthy and osteoarthritic samples were examined together as a complete chondrocyte dataset. We found that 90% of chondrocyte transcripts had <5 polyadenylation sites with 37% having just one isoform (Figure 2A). No transcripts were observed with more than 14 alternatively polyadenylated isoforms in our chondrocyte data set. We performed similar analysis on a human data set obtained from the polyA_DB database version 3.2 (https://exon.apps.wistar.org/polya_db/v3/, accessed on 21 September 2022), which is comprised of pooled polyadenylation site data from the analysis of 107 tissue and cell samples [19]. This polyA_DB data represents a reference of polyadenylation site usage across a wide variety of human cellular transcriptomes. When we compared chondrocyte polyadenylation site usage to that of the reference PolyA_DB data for 10,551 genes common to each dataset, it was evident that chondrocytes utilise a narrow band of the potential polyadenylation sites that have been so far defined in human cells and tissues (Figure 2B). Our chondrocyte data contained 1114 genes that exhibited >5 alternative polyadenylation sites. We performed ontological analysis of these highly alternatively polyadenylated chondrocyte genes to determine whether they were enriched for any specific characteristics. We found that for genes in chondrocytes that utilised this, many polyadenylation sites were strongly enriched for functions involved in mRNA processing and splicing as well as organelle formation and function (Figure 2C).

### 2.3. Characterisation of Highly Expressed, Multi-Polyadenylated Genes in Human Articular Chondrocytes

To examine the implication of alternative polyadenylation events in chondrocytes, we examined the ten most strongly expressed, multi polyadenylated genes across the healthy and osteoarthritic chondrocyte data (Table 1). We found that these genes were affected in varying ways based upon the polyadenylation site usage identified. Some genes exhibited simple truncations of the 3′UTR when more proximal sites were used (*DCN*, *SPARC*, *PRELP*, *CILP*, *EGR1*). Some of these genes (*DCN*, *PRELP*, *CILP*) did not always exhibit a distal polyadenylation site coincident with those present in the genomic annotation, indicating that in chondrocytes, all of their transcripts are truncated relative to these canonical sites. Other genes exhibited alternative polyadenylation that led to significant changes in mRNA structure. For instance, the chondrocyte data showed that two sites are used for *CFH*, the more proximal of which is a well characterised alternatively spliced variant that leads to a shorter protein-coding sequence. Less well characterised is the chondrocyte site usage for the *CLU* gene. Here, six sites were identified, three of which were close to existing annotated sites and were utilised in the majority of samples submitted to polyA_DB. Two of the other three sites were identified in a small proportion (<10%) of the samples used by polyA_DB. These three sites occur at the 3′ end of *CLU* exon 3 and represent around a quarter of all *CLU* reads in chondrocytes.

### 2.4. Osteoarthritis Does Not Result in Widespread Changes in Chondrocyte Polyadenylation Patterns

We used the expressRNA tool to determine the differential usage of polyadenylation sites between healthy and osteoarthritic cartilage samples. We found that disease-specific polyadenylation is not a widespread process with just 20 genes identified as being differentially regulated in this way in the diseased samples (where adjusted *p-*values for both distal and proximal sites relaxed to <0.1). These genes are presented in Table 2 with details of the alternative polyadenylation sites that were affected between healthy and osteoarthritic samples.

As expected, many of these APA events result in a truncated 3′UTR (KMT2A, SMAD5, SLC25A37, TMOD1) however there are examples of other events which would lead to more substantial changes to transcript structure. One of the most strongly regulated polyadenylation events was found for the lncRNA *NEAT1*. Healthy chondrocytes primarily use a distal *NEAT1* polyadenylation site, encoded at chr11:65,426,254, with a lower-level usage of sites 110 bp and 549 bp upstream of this.

Figure 3 presents the read data for three examples of genes where alternative polyadenylation was observed in osteoarthritis. Figure 3A shows gene structure diagrams for each of these genes, *OSMR*, *KMT2A* and *ANOS1*, with differential polyadenylation site location indicated with arrows. For the first example, we found that the use of a previously identified intronic polyadenylation site in the *OSMR* gene [20] was significantly downregulated in osteoarthritis (Figure 3B) There was also evidence of use of a further *OSMR* intronic polyadenylation site, 1.3 kb further downstream which was also diminished in use in osteoarthritic cells. For the *KMT2A* gene, osteoarthritis was associated with the reduced use of a proximal polyadenylation site in the 3′UTR (Figure 3C). Polyadenylation associated with the full-length mRNA for the *ANOS1* gene was only observed in osteoarthritic chondrocytes but there was consistent use of a site within intron 11 by control and osteoarthritic cells in all samples (Figure 3D). 

### 2.5. Intronic Polyadenylation of OSMR mRNA Leads to Altered Transcript Decay Rate and can Be Influenced by Inflammatory Cytokine Stimulation

Focussing firstly on examples of genes that exhibited intronic polyadenylation we were able to produce qPCR assays to assess expression of variants of the *OSMR* and *ANOS1* transcripts. We examined read counts for proximal and distal polyadenylation sites for these two genes and found that osteoarthritis led to a reduced use of the proximal, intronic site for *OSMR* whilst inducing the use of a distal site for *ANOS1* (Figure 4A). We next examined the expression and rates of mRNA decay of the isoforms of these two transcripts in osteoarthritic human chondrocytes (Figure 4B).

For the *OSMR* gene, the use of the intronic polyadenylation site, which results in a truncated protein-coding domain, also led to significant change in post-transcriptional regulation compared to the full-length transcript. The shorter transcript was less stable with a half-life of 9 h whereas the full-length transcript decayed slowly (half-life > 24 h). Interestingly, despite originating from osteoarthritic cartilage, the cultured chondrocytes expressed higher levels of the proximal isoform. For *ANOS1*, each isoform was very stable (>24 h half-life) and no difference was observed in their decay rates (Figure 4B). We examined whether the inflammatory cytokine IL-1ß, which is known to strongly regulate many osteoarthritis-associated genes, affected the use of the short and long isoforms of each gene in HACs (Figure 4C). We found that there was a significant reduction in the ratio of short to long isoform usage for *OSMR* over a 6 h exposure to IL-1ß, consistent with the shift in ratio observed between healthy and osteoarthritic tissue. The ratio of *ANOS1* isoforms was not significantly affected by the IL-1ß treatment.

### 2.6. mRNA Decay Is Affected by Polyadenylation Status of the KMT2A Transcript

Several transcripts that were differentially polyadenylated in osteoarthritis were characterised by differences in length of the 3′UTR. To analyse these isoforms with qPCR we were only able to specifically identify longer, distally polyadenylated isoforms because they contain a specific region of 3′UTR sequence. We therefore intended to use ratios of “distal” to “proximal and distal” site usage to allow us to differentiate between APA isoforms and allow us to determine how they affected transcript decay rates. 

For many targets, designing discriminatory assays with satisfactory amplification efficiencies was challenging. We were, however, able to identify primer pairs with comparable amplification efficiencies that could be used to analyse of the *KMT2A* and *SLC25A37* isoforms. Each of these genes exhibited significantly reduced use of a proximal polyadenylation site in osteoarthritic samples (Figure 5A). Using qPCR, we found it difficult to differentiate between expression levels of proximal and distal isoforms for the genes *KMT2A* and *SLC25A37* in both HAC and SW1353 cells (Figure 5B). Because these were genes that exhibited lower proximal site usage in OA this indicated that in cultured human chondrocytes the predominant isoform uses the distal polyadenylation site. Unsurprisingly, the results of actinomycin D chase experiments in chondrocytes led to similar results with each primer pair, as the distal isoform was likely to be predominately detected. We still wanted to understand how the APA of these genes might affect mRNA decay so we screened a range of cell types of different tissue origins in an attempt to identify a model system where a higher ratio of short and long isoforms were expressed (Figure 5C). For *SLC25A37*, we found that a ratio of 1 in qPCR products was generally consistent across all of the cell types we analysed, again indicating mostly proximal site usage. However, for *KMT2A* we found that two of the cell lines screened, AC10 and MDA-MB-231, exhibited short to long isoform ratios of 2 and above indicating that they were also expressing the proximally polyadenylated form. We also measured APA usage for *OSMR* and *ANOS1* in the cell lines to determine whether there was any common regulation of the osteoarthritic APA sites across different cell types. This did not seem to be the case: *ANOS1* chiefly exhibited preponderance to proximal site usage across all of the lines. In contrast, *OSMR* site usage varied between cell types, with A549 and NTMS cells making less use of the intronic proximal site compared to the other four cell lines. Based upon our cell-line data, we chose to use the AC10 cells to examine whether APA affected mRNA decay rates for *KMT2A* (Figure 5D). Using our proximal and distal qPCR assays we found that the product of the distal, long-isoform-specific PCR was more stable than the product of the proximal PCR which detects both long and short isoforms. We can infer from this that the shorter *KMT2A* transcript is less stable than the longer one.

## 3. Discussion

This is the first study to examine in detail the use of polyadenylation sites by articular chondrocytes and the effect of osteoarthritis on this process. It has confirmed that chondrocytes make widespread use of APA to create heterogeneity in their transcriptomes. However, change in the pattern of APA is not a widespread feature of the transcriptomes of osteoarthritic chondrocytes.

QuantSeq-Reverse leads to 3′ end reads only and is particularly well suited to studying polyadenylation patterns because each read sequences from the polyA tail into the 3′UTR of a transcript. This approach makes the technique less efficient than other 3′ sequencing approaches for transcript expression profiling but does not preclude such analyses from being conducted. Encouragingly, we have found that our dataset allowed us to identify many transcripts with altered abundance between osteoarthritic and age-matched normal cartilage tissue. Around a third of the highly regulated genes that we have observed have been similarly identified in another transcriptional profiling study examining the disease [5].

In line with the nature of osteoarthritis as a disease, ontology analysis of differentially regulated genes showed enrichment for processes associated with the skeleton, joints and connective tissue; areas that are all known to be significantly affected by osteoarthritis. Our use of age-matched human control and diseased samples allowed us to build on previous transcriptomic studies. Currently, our study and that of [5] are the only transcriptomics studies that make use of age-matched healthy control tissue for the study of human osteoarthritis. Other studies have examined transcriptome or epigenome alterations between intact and fibrillated tissue from osteoarthritic joints [21,22] and have observed differences in expression but our data and that described by Soul and colleagues in [5] indicate that this comparison may not be fully reflective of the differences between diseased and healthy tissue. This approach may have value, because control-tissue choice represents a continual obstacle in the study of human osteoarthritis, where appropriately matched, non-diseased human cartilage can be challenging to obtain. However, an acceptance of potential limitations of different approaches is required. Our healthy samples were sourced from a commercial supplier, whilst we collected osteoarthritic tissue from a local hospital following knee arthroplasty procedures. PCA analysis of the transcriptome data showed that there was significantly less variance in the osteoarthritic samples compared with the healthy, a finding that we have observed in other RNA-sequencing studies [23]. This may reflect a greater homogeny of phenotype among late-stage osteoarthritic samples, in line with a growing opinion that differing age related mechanisms can converge into a common end stage form of osteoarthritis [24].

A characteristic property of the QuantSeq-Reverse technique is that sequences originate within the polyA tail and proceed 3′–5′ into the transcript, meaning that every read specifically identifies a transcript polyadenylation site. This has allowed for fresh insights into transcript regulation in chondrocytes at the post-transcriptional level by identifying patterns of chondrocyte polyadenylation site usage as well as determining the extent of alternative polyadenylation that occurred between age-matched healthy and osteoarthritic cartilage samples. We confirmed that chondrocytes make widespread use of alternative polyadenylation to enhance the diversity of their transcript pool although the number of sites used is substantially lower than those identified across multiple human tissue and cell types in the polyA_DB database [25]. It is therefore likely that the chondrocyte employs a specific pattern of polyadenylation usage, some aspects of which contribute to phenotype-specific gene functionality. An example of this would be the clusterin gene where a quarter of the chondrocytes’ reads indicate relatively high use of polyadenylation sites towards the 3′ end of exon 3, which have been previously identified in this region but are not commonly observed in other cell types. Exon 3 terminating transcripts have been annotated as *CLU-204* and *CLU-214* and are predicted to code a truncated protein, although they would be polyadenylated directly on the protein-coding region with no 3′UTR. The polyadenylation sites identified in the chondrocytes would actually lead to transcripts between 20 to 83 bp shorter again than those annotations. It is currently not understood what the function of these truncated clusterin transcripts is or whether they are even translated.

It was instructive that chondrocytes from osteoarthritic cartilage do not exhibit widespread altered usage of polyadenylation sites, indicating that a systematic change in the processes controlling APA is not a significant feature of the disease. This distinguishes the processes occurring in osteoarthritis from other pathologies. For instance, many cancers exhibit global regulation of alternative polyadenylation events [15,16]. Despite osteoarthritis lacking these widespread changes, a small number of alternatively polyadenylated transcripts emerged from our analysis and potentially have a capacity to contribute to the disease mechanism. One of the most interesting is the oncostatin-M receptor *OSMR*. The use of an intronic polyadenylation site has already been demonstrated for this gene, which results in a truncated, secreted form of the receptor that can act as an oncostatin-M antagonist [20]. We have shown here that the use of the intronic site is decreased in osteoarthritic cartilage, which is likely to lead to reduced levels of this antagonistic protein isoform and increased oncostatin-M signalling in osteoarthritic tissue. It was interesting to note that stimulation of chondrocytes with IL-1ß led to a decrease in the ratio of polyadenylation usage at the intronic site versus the full-length form, indicating that inflammatory conditions are able to promote a similar reduction in levels of the oncostatin-M antagonistic form. Inflammatory cytokine stimulation can promote many of the catabolic processes that lead to extracellular matrix destruction in osteoarthritis and oncostatin-M has recently been associated with inflammatory subtypes of osteoarthritis [26]. Understanding how loss of oncostatin-M antagonism through altered use of the regulated intronic polyadenylation sites that we have observed might contribute to disease state warrants further investigation.

The gene *KMT2A* encodes a lysine methyl transferase, and its expression was downregulated overall in osteoarthritic tissue. It has previously been linked to a potential causal role in altered cartilage thickness through genome-wide association studies [27]. It also exhibited an almost complete absence of the use of a proximal polyadenylation site within its 3′UTR in osteoarthritis, which could influence the dynamics of post-transcriptional processing. Because our chondrocyte primary cell models are from osteoarthritic tissue, we were unable to identify whether this alteration in 3′UTR length affected mRNA decay because it appears that the cells only express the longer version. However, following a screen of a variety of cell lines we found that AC10 cells appear to produce both long and shorter isoforms of the *KMT2A* transcript and that shortening of its 3′UTR leads to a less stable mRNA. This polyadenylation event may therefore play an important role in maintaining KMT2A levels in healthy cartilage and it would be useful to elucidate the mechanisms that lead to its use being suppressed in osteoarthritis.

Long noncoding RNAs have been increasingly studied in relation to osteoarthritis recently and DeSeq2 analysis of the QuantSeq-Reverse data has identified 18 significantly regulated lncRNAs that had not previously been associated with joint pathology. Furthermore, an alternative polyadenylation event was identified for the lncRNA *NEAT1*, which has been extensively studied in a variety of tissues and has been found to function as microRNA sponge chondrocytes [28]. The functional significance of this 110 bp difference is not understood although the sequence in this region is more strongly conserved than the rest of the 3′ half of most *NEAT1* transcript isoforms. A limitation of the QuantSeq-Reverse approach appears to have been a poor coverage of small non-coding RNAs such as microRNAs and snoRNAs. We did not specifically prepare our RNA samples to enrich for small RNAs, which may explain this. These molecules are known to perform important regulatory roles in chondrocytes and can be regulated in osteoarthritis. They are polyadenylated and so our inability to fully characterise them in our samples represents a limitation of the study. Interestingly, among the small RNAs that we were able to detect were potential biomarkers for osteoarthritis, such as *miR-320d* [29] and *SNORD13* [30].

Our study has been able to identify only a low level of APA caused by osteoarthritis but is subject to some other limitations. Firstly, the number of replicates did not allow us to stratify patients by comorbidities and meant that we limited our study to samples from males, meaning that we have not been able to define any gender-specific effects of APA. Therefore, we may have missed some polyadenylation events that are sensitive to these parameters. We may have also not have been able to effectively quantify genes that were expressed at low levels, which may have led to an underestimation of significant APA in osteoarthritis. Further work is also required to better understand the significance of APA events. We have some evidence that mRNA decay can be regulated by APA for *KMT2A* but have not looked at other parameters such as translation efficiency and subcellular localisation of the transcripts in this study.

Overall, this study has demonstrated that altered polyadenylation site usage is not a widespread characteristic of osteoarthritis, but that a small number of genes are affected, and the implications of the changes could have significant potential to affect their functions. Further understanding of how to control polyadenylation in chondrocytes across the whole transcriptome is therefore unlikely to be required to combat osteoarthritis but influencing specific polyadenylation events, the loss of intronic polyadenylation of OSMR for instance, could still provide important targets for the modulation of this disease.

## 4. Materials and Methods

### 4.1. Age-Matched Healthy and Osteoarthritic Articular Cartilage

Osteoarthritic human articular cartilage tissue was obtained from the knee joints of patients undergoing total knee arthroplasty (Clatterbridge Hospital, Wirral, UK) due to late-stage disease. This material was obtained with informed consent and following national research ethics committee approval (IRAS ID 242434). Age-matched, control human articular cartilage tissue was obtained commercially (Articular Engineering, Northbrook, IL, USA). All control samples had been obtained post-mortem (within 24 h of death) from individuals with no record of joint disease whose joints were macroscopically healthy on visual inspection. Six cartilage samples from each group were analysed and all were obtained from male donors. All tissue was harvested from load-bearing regions of the femoral condyle. Samples were snap-frozen and 10 µm thick full-depth sections of the tissue were cut using a cryostat and collected prior to RNA isolation.

### 4.2. QuantSeq-Reverse Analysis

Total RNA was isolated from tissue cryosections, to ensure consistent use of full depth tissue, using a Trizol extraction (Thermo Fisher Scientific, Horsham, UK) with subsequent purification using the RNeasy spin column (Qiagen, Manchester, UK) purification procedure including on-column DNase digestion [31]. QuantSeq-Reverse library preparation and NextSeq 75 cycle High Output sequencing with V2chemistry, was performed by Lexogen (Vienna, Austria) [18]. For gene-expression analysis, the in-house Lexogen pipeline was used, consisting of read-trimming, alignment to GRCh38 using STAR aligner [32] and then differential gene expression determined using DESeq2 [33]. Correction for multiple hypothesis testing was performed via the Benjamini–Hochberg procedure [34]. Differentially expressed genes, defined as log2 fold change ≥ 1 and ≤−1 and adjusted *p*-values (≤0.05), were used to distinguish between downregulated and upregulated genes. Raw FastQ files from QuantSeq-Reverse analysis have been deposited in ArrayExpress (https://www.ebi.ac.uk/arrayexpress/) on 1 October 2022 with the accession number E-MTAB-12184.

### 4.3. Alternative Polyadenylayion Analysis

We constructed a reference polyA database by considering genome alignments of sequence reads from all twelve experiments. To account for internal-priming events (reads mapping to genomic polyA sequences) we filtered out detected polyA sites without an upstream polyadenylation signal (PAS). After obtaining read coverage for each detected polyA site separately for each experiment, we ran DEXSeq analysis comparing six healthy vs. six osteoarthritis samples to identify alternatively polyadenylated (APA) genes. Further polyadenylation analysis details and methodology can be explored at expressRNA [35].

### 4.4. Gene Ontology and Pathway Analysis

Enriched KEGG [36] and Reactome [37] pathways were identified using the clusterProfiler package [38] and visualised using the pathview package [39]. For gene-set enrichment analysis (GSEA), a pre-ordered list of gene-named log2 fold change values, and for over-representation analysis (ORA), an input list of differentially expressed genes was used to obtain enriched pathways. Upregulated pathways were defined by a normalised enrichment score (NES) > 0 and the downregulated pathways were defined by an NES < 0. Pathways with BH adjusted *p-*value ≤ 0.05 were chosen as significantly enriched. RcisTarget v1.9.1 [40] was applied on differentially expressed genes to identify overrepresented potential transcription factor regulation. We examined up to 500 bp upstream of the transcription start site, and TF_highConf (TFs, annotated to the motif according to ‘motifAnnot_highConfCat’ of Rcistarget) were considered associated with TFs of interest.

### 4.5. Human Articular Chondrocyte Isolation

Osteoarthritic tissue samples were obtained as described above. Chondrocytes were isolated from finely diced cartilage tissue by digestion in 0.08% collagenase type II (Worthington Biochemical, Lakewood, NJ, USA) in growth medium (Dulbecco’s modified Eagle’s medium, containing 10% foetal bovine serum, 100 units/mL penicillin and 100 units/mL streptomycin, all from Thermo Fisher Scientific) overnight at 37 °C with agitation. Isolated cells were washed three times in growth medium without collagenase, plated at 20,000 cells/cm^2^, and grown for 2 passages at 37 °C in 5% CO_2_ with 1:2 split ratios before use in experiments.

### 4.6. Cell Culture

For analysis of the ratios of alternatively polyadenylated transcripts the following cell types were screened: SW1353 chondrosarcoma, AC10 ventricular cardiomyocytes, MG63 osteosarcoma, NTM5 trabecular network cells, A549 lung carcinoma and MDA-MB231 breast adenocarcinoma cells. All cell lines were cultured in growth medium at 37 °C in 5% CO_2_. For mRNA decay analysis, confluent cell monolayer cultures of the cell lines of interest were cultured in standard media supplemented with 1 ng/mL actinomycin for 0, 2, 4, 6 and 8 h before lysis using TRIzol reagent. Purification of cultured cell total RNA was performed by chloroform-based phase separation and subsequent precipitation with isopropanol. RNA was redissolved in RNAse free H_2_O and stored at −80 °C until required [9]. Cytokine stimulation of human articular chondrocytes was performed on cells at confluence by adding 10 ng/mL interleukin-1β (IL-1β, Peprotech, London, UK) to standard media and stimulating cells for up to 6 h.

### 4.7. qRT-PCR Analysis

cDNA was synthesized using 1μg total RNA with 200 units Moloney murine leukemia virus reverse transcriptase (Promega, Southampton, UK), primed with 2 μM oligo dT (Thermo Fisher Scientific). qPCR was performed on a Roche Lightcycler 96 using Takyon ROX SYBR MasterMix (Eurogentec, Seraing, Belgium) using transcript-specific primer pairs at a concentration of 300 nM per primer. Primer efficiency for each assay was established as previously described and assays were only used where efficiencies were within 10% of those of the reference gene [41]. Primer sequences for each assay can be found in Table 3. The 2^−ΔCt^ method was used to calculate gene-expression levels [42] with *RPS13* used as the reference expression gene [43].

### 4.8. Statistical Analysis

Where appropriate, analysis of cell culture experiments was conducted using one-way analysis of variance as described in figure legends and a *p-*value < 0.05 was considered significant. Statistical methodology underpinning DeSeq2 and expressRNA analysis of transcriptomic datasets has been described previously [33,35].

## Figures and Tables

**Figure 1 ijms-24-06611-f001:**
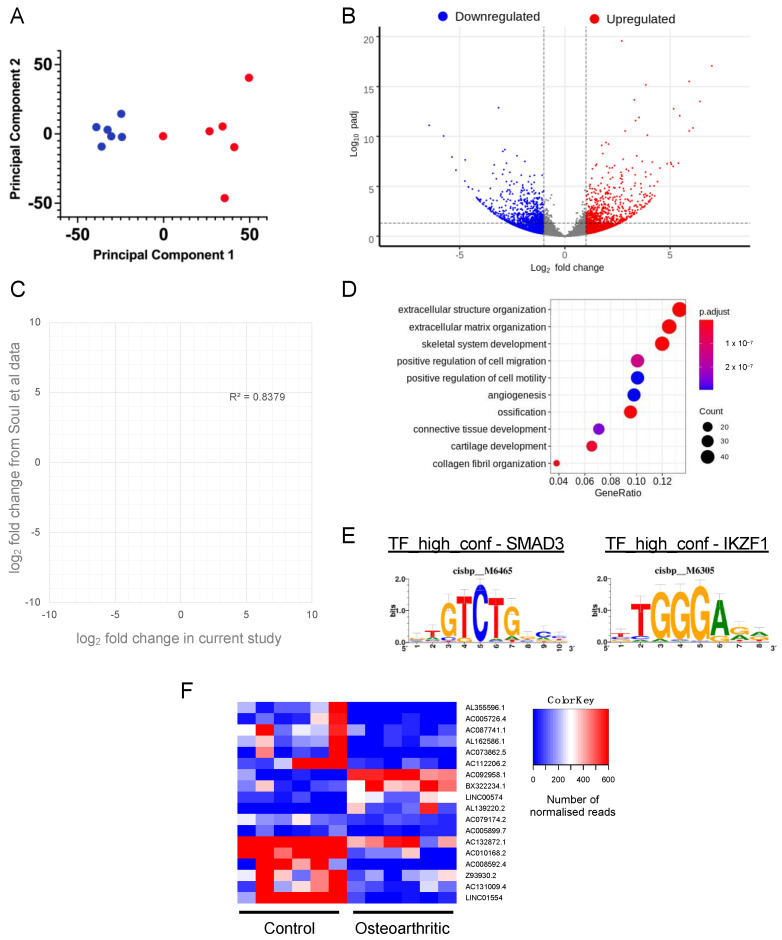
(**A**) Principal component analysis following determination of differential gene expression in QuantSeq-Reverse dataset using DeSeq 2. Red points are control samples and blue points are osteoarthritic samples. (**B**) Volcano plot showing differential expression of genes between datasets, with genes significantly upregulated in osteoarthritis highlighted in red and genes significantly downregulated in osteoarthritis highlighted in blue. (**C**) Scatterplot showing the relationship between the fold changes of genes significantly affected by osteoarthritis in this study and also significantly identified between intact and degenerate osteoarthritic cartilage in a previous study by Soul et al. [5]. The dashed line and R^2^ value are the result of linear regression analysis. (**D**) Gene ontology analysis of genes that were identified as being differentially regulated in osteoarthritic samples. The colour of each circle represents the *p-*value associated with the enrichment whilst circle size is associated with the number of genes associated with ontological term. (**E**) Transcription factor-binding motifs enriched in the promoters of genes that are differentially regulated by osteoarthritis. (**F**) Heatmap illustrating the expression of lncRNAs that are differentially expressed in osteoarthritic samples compared to controls.

**Figure 2 ijms-24-06611-f002:**
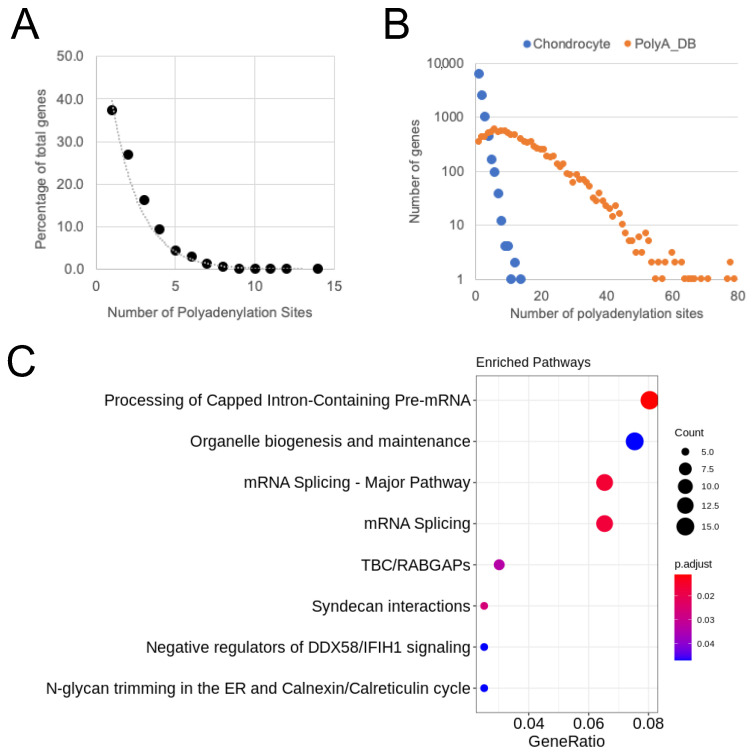
(**A**) The number of cartilage genes exhibiting different numbers of polyadenylation sites plotted as a percentage all genes. (**B**) Counts for the number of genes with different numbers of polyadenylation sites identified across control and osteoarthritis cartilage samples combined (blue circles) and human data from polyA_DB (orange circles). (**C**) Gene ontology analysis of genes where >5 polyadenylation sites were identified across all the cartilage samples analysed. The colour of each circle represents the *p-*value associated with the enrichment whilst circle size is associated with the number of genes associated with the ontological term.

**Figure 3 ijms-24-06611-f003:**
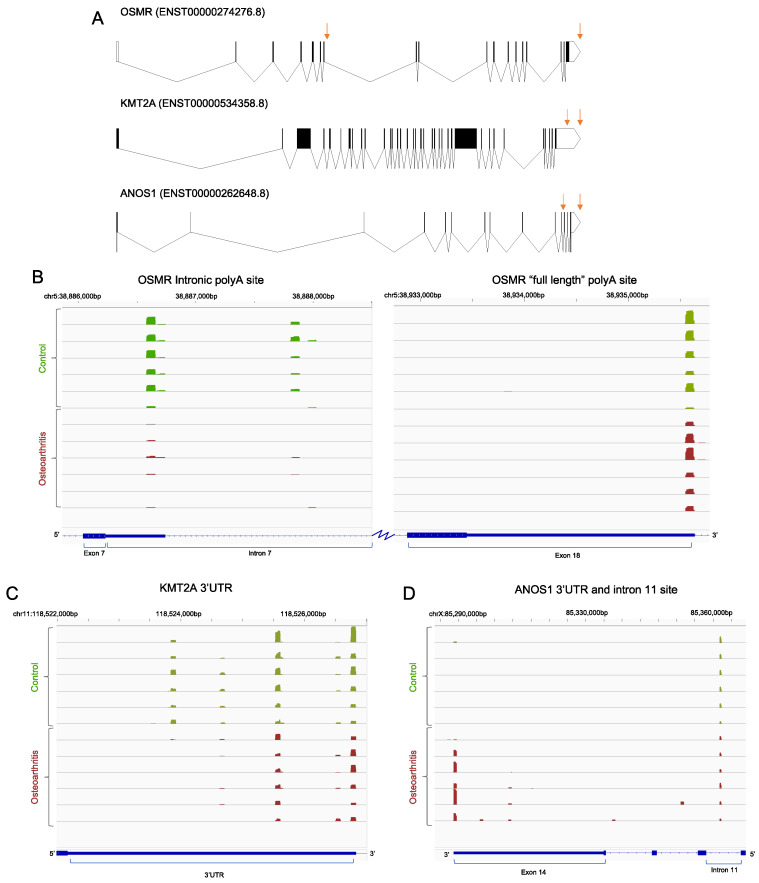
(**A**) Gene structure diagrams for *OSMR*, *KMT2A* and *ANOS1*. ENSEMBL IDs for the specific transcripts shown are indicated in parenthesis. Blocks represent exons and lines represent introns. Black shaded exon regions represent the coding domain. Orange arrows indicate the locations of polyadenylation sites that are differentially utilised in osteoarthritis. Note that the diagrams in this panel are not presented to scale. (**B**–**D**) Genome browser track example showing QuantSeq Reverse reads for three genes affected by alternative polyadenylation in osteoarthritic chondrocytes. Tracks for the six control (green) and six osteoarthritic (red) samples are shown. (**B**) Reads containing *OSMR* polyadenylation sites at intron 7 (left panel), which will lead to a transcript encoding a secreted form of OSMR and at the full-length transcript’s 3′UTR (right panel) which would encode the full plasma membrane receptor transcripts (each track is scaled from 0–3600 reads). (**C**) Polyadenylation site reads in the 3′ UTR of *KMT2A* transcripts (each track is scaled from 0–691 reads). (**D**) Reads containing intronic and full length 3′UTR polyadenylation sites for ANOS1 transcripts (each track is scaled from 0–590 reads). Note that *OSMR* and *KMT2A* are encoded on the sense strand and so are presented 5′–3′ left to right in panels B and C. *ANOS1* is encoded on the antisense strand and so runs 5′ to 3′ right to left in panel D.

**Figure 4 ijms-24-06611-f004:**
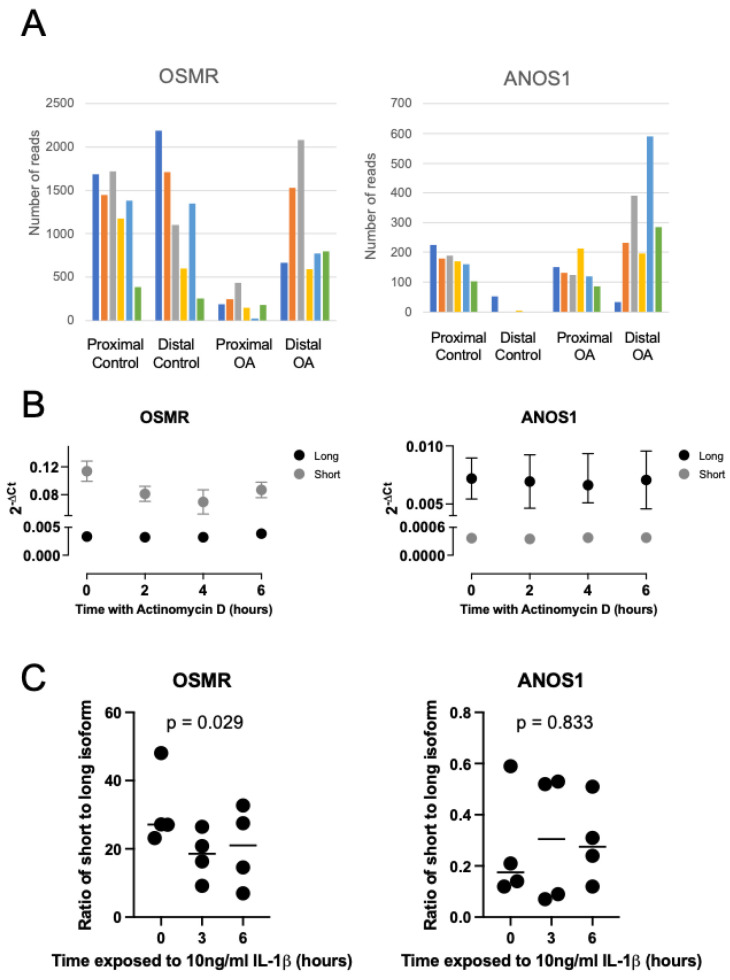
(**A**) QuantSeq-Reverse read numbers for proximal (intronic) and distal (3′UTR) polyadenylation sites for the *OSMR* and *ANOS1* genes, which were differentially regulated between healthy (control) and osteoarthritic (OA) cartilage samples. *OSMR* proximal site is at CHR5:38886653, *OSMR* distal site is at CHR5:38935628, *ANOS1* proximal site is at CHRX:8,536,187, ANOS1 distal site is at CHRX:85,28,874. Different colour bars represent read numbers from individual control or OA samples allowing per sample comparison between proximal and distal values in each case. (**B**) RT-qPCR analysis of mRNA decay rates of *OSMR* and *ANOS1* transcripts that use intronic or canonical 3′UTR polyadenylation sites in human articular chondrocytes treated with actinomycin D. Mean ± standard error shown, n = 3. (**C**) RT-qPCR analysis of the ratio of intronic to canonical 3′UTR transcript levels of *OSMR* and *ANOS1* in HAC treated with 10 ng/mL interleukin-1ß (IL-1ß). Individual data points are presented (n = 4) whilst line represents mean value and *p-*values represent result of one-way analysis of variance.

**Figure 5 ijms-24-06611-f005:**
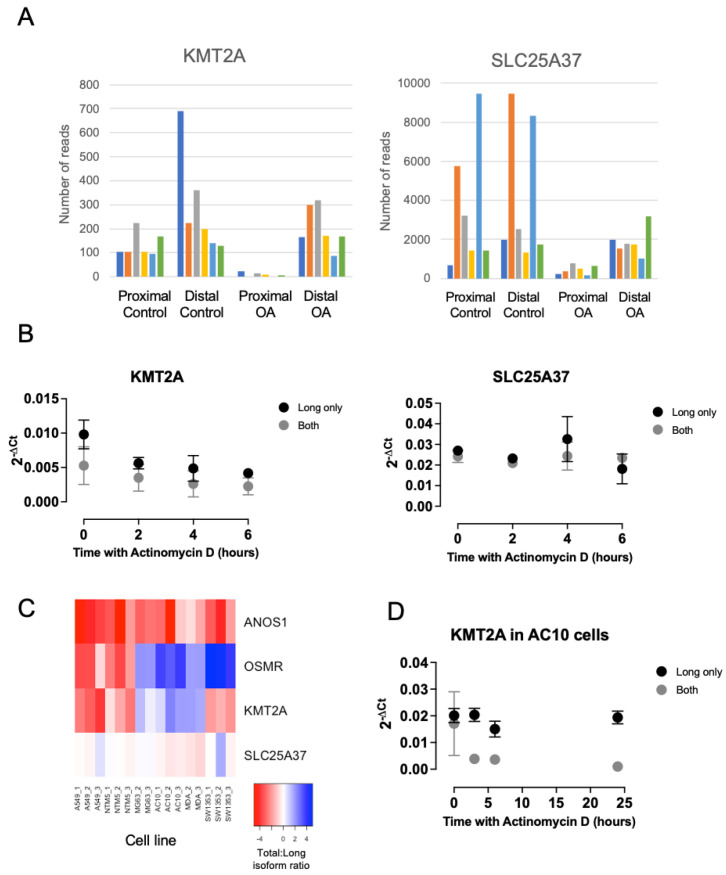
(**A**) QuantSeq-Reverse read numbers for proximal and distal polyadenylation sites for the *KMT2A* and *SLC25A37* genes, usage of which leads to different length 3′UTRs and is differentially regulated between control and osteoarthritic (OA) samples. *KMT2A* proximal site is at CHR11:118523915, *KMT2A* distal site is at CHR11:118526830, *SLC25A37* proximal site is at CHR8:23,572,547, *SLC25A37* distal site is at CHR8:235,75,460. Different colour bars represent read numbers from individual control or OA samples allowing per sample comparison between proximal and distal values in each case. (**B**) RT-qPCR analysis of mRNA decay rates of *KMT2A* and *SLC25A37* using primer pairs that are specific to only longer 3′UTR isoforms (long only) or both proximal and ideally polyadenylated isoforms (both) in human articular chondrocytes treated with actinomycin D. Mean ± standard error shown, n = 3. (**C**) Heat map showing ratio of expression of intronic to canonical 3′UTR (for *OSMR* and *ANOS*1) or long only to both for (*KMT2A* and *SLC25A37*) in a panel of different cell lines. (**D**) RT-qPCR analysis of mRNA decay rates of long only or both polyadenylated isoforms of *KMT2A* in AC10 cells. Mean ± standard error shown, n = 3.

**Table 1 ijms-24-06611-t001:** The 10 most highly expressed alternatively polyadenylated genes in articular chondrocytes regardless of disease state.

Gene Name, (Gene Symbol), Ensembl Gene ID	Gene Locus (Strand)	Sites (Proximal to Distal)	Site Distance from Proximal Site (bp)	Site Mean Read Count ± SD
Decorin, (*DCN*), ENSG00000011465	chr12:91140483–91182823 (antisense)	91145968	0	12,270 ± 8494
		91145870	98	19,760 ± 13,585
		91145526	442	51,032 ± 39,570
Noncoding nuclear-enriched abundant transcript 2, (*MALAT1*), ENSG00000251562	chr11:65497687–65506515 (sense)	65499339	0	28,517 ± 21,758
		65499659	320	21,804 ± 17,950
		65499698	359	100,707 ± 98,832
		65499736	397	99,063 ± 96,843
		65499776	437	50,554 ± 37,148
		65499794	455	5531 ± 4083
		65500522	1183	6120 ± 6761
		65500558	1219	5446 ± 5383
Complement factor H, (*CFH*), ENSG00000000971	chr11:65497687–65506515 (sense)	196701562	0	38,800 ± 68,040
		196747501	45,939	8984 ± 9034
Clusterin, (*CLU*), ENSG00000120885	chr8:27596916–27614699 (antisense)	27609020	0	2143 ± 1472
		27608995	25	6215 ± 3978
		27608957	63	7438 ± 4695
		27597996	11,024	33,471 ± 22,377
		27597813	11,207	8339 ± 9719
		27597728	11,292	21,976 ± 21,562
Osteonectin, (*SPARC*), ENSG00000113140	chr5:151661095–151686974 (antisense)	151662435	0	22,006 ± 29,354
		151661449	986	4341 ± 5672
60S ribosomal protein L7a, (RPL7A), ENSG00000280858	chr9:133348217–133351425 (Sense)	133351412	0	4060 ± 2684
		133351424	12	18,479 ± 12,447
Prolargin, (*PRELP*), ENSG00000188783	chr1:203475805–203491351 (sense)	203487161	0	17,465 ± 13,603
		203489613	2452	16,419 ± 13,500
Cartilage intermediate layer protein, (*CILP*), ENSG00000138615	chr15:65194759–65211472 (antisense)	65196529	0	3135 ± 4161
		65196251	278	13,156 ± 16,572
		65196000	529	4965 ± 5868
H3 histone family member 3B, (*H3F3B*), ENSG00000132475	chr17:75776433–75785892 (antisense)	75778250	0	2376 ± 2438
		75778023	227	2255 ± 2841
		75777456	794	12,733 ± 12,977
Early growth response 1, (*EGR1*), ENSG00000120738	chr5:138465478–138469302 (sense)	138469224	0	2641 ±2780
		138469240	16	729 ± 649
		138469301	77	12,484 ± 16,106

**Table 2 ijms-24-06611-t002:** Genes affected by alternative polyadenylation in osteoarthritic chondrocytes with corrected *p-*values for distal and proximal sites set to <0.1.

Gene Locus	Strand	ENSEMBL Gene_id	Gene Symbol	Proximal/Distal polyA Sites	Proximal/ Distal Feature	Proximal log2fc	Distal log2fc	Proximal Adjusted *p*-Value	Distal Adjusted *p*-Value
chr11:118482737–118526831	+	ENSG00000118058	*KMT2A*	118523915, 118526830	utr3/utr3	2.06168	−1.67404	0.01625	0.00558
chr11:10812073–10858795	-	ENSG00000236287	*ZBED5*	10852797, 10852747	utr3/utr3	2.29148	−2.94875	0.01812	0.0418
chr9:97501179–97601742	+	ENSG00000136842	*TMOD1*	97600155, 97601252	utr3/utr3	1.51617	−1.24927	0.02117	0.01554
chr21:15730024–15880068	+	ENSG00000155313	*USP25*	15878905, 15880062	utr3/utr3	−1.52865	3.44074	0.00356	0.054
chr11:65422773–65445539	+	ENSG00000245532	*NEAT1*	65426414, 65426524	exon/exon	−0.82173	1.04492	0.06951	0.08359
chr19:45349836–45370917	-	ENSG00000104884	*ERCC2*	45351594, 45349837	utr3/utr3	−1.70459	2.32543	0.0418	0.07641
chr11:111540279–111561744	+	ENSG00000204381	*LAYN*	111560953, 111561510	utr3/utr3	−1.46532	2.04648	0.06024	0.09604
chr5:136132844–136188746	+	ENSG00000113658	*SMAD5*	136179501, 136182732	utr3/utr3	−3.68803	1.67826	0.08576	0.00919
chr9:129634603–129642168	-	ENSG00000148331	*ASB6*	129636893, 129634604	utr3/utr3	−1.45783	1.29424	0.09604	0.08311
chr12:51186935–51217707	-	ENSG00000184271	*POU6F1*	51189184, 51186936	utr3/utr3	1.70409	−1.32788	0.08359	0.0527
chr8:23528955–23575462	+	ENSG00000147454	*SLC25A37*	23572547, 23575460	utr3/utr3	0.68296	−1.24319	0.06512	0.07641
chr2:44316280–44361861	-	ENSG00000138078	*PREPL*	44318763, 44317605	utr3/intron	−1.54952	2.27364	0.01646	0.04608
chr5:38845857–38945595	+	ENSG00000145623	*OSMR*	38886653, 38935628	intron/utr3	0.7451	−1.81506	0.03228	0.00233
chr5:132556018–132646348	+	ENSG00000113522	*RAD50*	132616045, 132642473	utr3/utr3	3.07214	−5.49643	0.00683	0.04908
chr3:48241099–48299252	+	ENSG00000164048	*ZNF589*	48270986, 48273225	utr3/intron	−0.70227	0.75939	0.04908	0.05682
chr7:77798791–77957502	+	ENSG00000006576	*PHTF2*	77940457, 77957223	utr3/utr3	3.12642	−2.27133	0.05189	0.01812
chr2:85751343–85791382	+	ENSG00000168874	*ATOH8*	85775209, 85791376	exon/utr3	3.02271	−3.80491	0.01812	0.05189
chr6:85449583–85495790	+	ENSG00000135318	*NT5E*	85452154, 85495783	intron/utr3	1.14747	−1.70338	0.04908	0.0273
chrX:8528873–8732136	-	ENSG00000011201	*ANOS1*	8536187, 8528874	intron/utr3	3.04985	−3.07918	0.00008	0.00008
chr12:19404044–19720800	+	ENSG00000139154	*AEBP2*	19441779, 19520222	intron/utr3	3.11654	−1.15156	0.09497	0.00558

**Table 3 ijms-24-06611-t003:** Sequences of primers used in qRT-PCR assays.

Assay Name	Forward Primer Sequence (5′–3′)	Reverse Primer Sequence (5′–3′)
RPS13	GGCTTTACCCTATCGACGCA	GTCAGATGTCAACTTCAACCAAGTG
OSMR proximal	GCAGTAGGTTGTCTGGGTCA	TGAGCCTGGAACAAACAGCA
OSMR distal	TCATTCACAGCGGAGGTGAG	GCGTGCATCCATGAGGAGAA
ANOS1 proximal	CCAGATCCTGCCTTCCGTAG	AGTGCAGCACACAGAATGAC
ANOS1 distal	ATGCAGATGCCTGGCCATT	AGGCTCTGTGGAGTACAGTGA
KMT2A proximal	CACCTACAGCGTCTGTCGAA	CACCGGAGGTGCTAGGAATC
KMT2A distal	GGGGTTCCACTAGTGTCTGC	TTCAGGACCCATCAGTGCAT
SLC25A37 proximal	GGTGCATCTTACCGAGGAGG	GTGCATGGTCCCAAGATGGA
SLC25A37 distal	GTGTGCTTGTGCGTGTCTAC	TGTTGCCTTTTCGTTCACCTG

## Data Availability

Raw FastQ files from QuantSeq-Reverse analysis have been deposited in ArrayExpress (https://www.ebi.ac.uk/arrayexpress/) on 1 October 2022 with the accession number E-MTAB-12184.

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
