# Peer review of "Analysis of RNA Polyadenylation in Healthy and Osteoarthritic Human Articular Cartilage"

_ijms, 2023, doi:10.3390/ijms24076611_

Round 1

Reviewer 1 Report

In the manuscript, Winstanley-Zarach et al tried to explore the distribution of polyA sites on articular chondrocyte transcripts in the health and OA cartilage. They found that differential use of polyadenylation sites was modest. Their findings could provide new insight into OA study. Several issues should be addressed in the study.

(1) For the OA samples used in the study, which stage of OA are they from? Early stage? Moderate stage? Or Late stage? The gene expression is different by stages.

(2)For the QuantSeq Reverse analysis, total RNA was isolated from tissue cryosections. Why not try to directly isolate RNA from fresh tissue?

(3) For the culture of human chondrocytes, is it normal culture with oxygen? Because cartilage is in the hypoxia environment, chondrocytes typically have a adapted response to low oxygen environment. Why not use the hypoxia culture for chondrocytes?

(4) Statistical analysis is critical for a study. Could the author indicate the statistical analysis as a sub-section in the method?

(5) Could the author clarify the application in OA? Which one could be therapeutic target for OA in the future?

Reviewer 2 Report

Analysis of RNA polyadenylation in healthy and osteoarthritic human articular cartilage described by Phaedra Winstanley-Zarach et al. is a very interesting paper. However, it is insufficient for publication in several respects.

First, numerous papers have been reported on the regulation of polyadenylation and its involvement in various life events and diseases such as cancer and development. However, in the Introduction section, the authors present few previous reports. Also, there is a lack of explanation as to why they decided to analyze APA for cartilage in OA.

In the pathogenesis of OA, it is more likely that long-term physical loading and aging are responsible for the onset of OA, rather than gene expression changes in cartilage. Therefore, the authors should clearly state that this study tested whether mRNA modifications are altered in the pathogenesis.

Regarding the fact that there were only 20 genes with changes in polyadenylation shown in Table 2, it should be made clear in the database-derived experimental data that this is not due to mere individual differences.

The lack of description of cartilage samples does not guarantee the reproducibility of this study. For example, the patient's race, height, weight, BMI, disease grade, presence of synovial fluid retention, and course of treatment up to sampling should be specifically described. OA is a disease that is diagnosed and treated according to the patient's complaints. Therefore, even if radiographs show obvious abnormalities in the knee cartilage, there may be cases in which the patient does not necessarily experience severe pain. For this reason, the absence of abnormalities in the knee cartilage of a healthy person should be described based on objective clinical data, not the complaint.

Regarding changes in polyadenylation, for representative genes (OSMR, KMT2A, etc.), it will help the reader's understanding if the gene structures is illustrated.

For genes with low expression, the possibility of missing them due to low read counts should be explained.

The significance of the study is unclear because the potential for better treatment and diagnosis is not well-described.

It should describe whether the various mRNA variants detected are in the cell nucleus or in the cytoplasm, taking into account mRNA degradation pathways due to nonsense mutations, etc.

Reviewer 3 Report

The authors studied alternative polyadenylation (APA) in human knee osteoarthritis using the QuantSeq Reverse approach. They provided a comprehensive mapping of polyadenylation site usage in chondrocytes and the differentially expressed transcripts in OA vs healthy control. The authors found that APA is not likely very altered in OA. However, they applied relatively precious non-OA samples as a control, the data set is helpful for studying OA.

Overall, the experiments are well-designed and executed, and the findings are interesting and novel. However, before the manuscript can be published, the authors should address the following issues.

  1. In figure 1, authors should be consistent in their use of color, e.g., red dots represent OA samples (A), red dots are for genes that are up-regulated in OA (B) and red is as the higher value in the heatmap (E). Confirm again in the additional table that the logFC values correspond to the correct compare direction (OA vs Con, or Con vs OA). We recommend the former, OA vs Con. Please also include the raw counts or normalized counts to Table S1.
  2. Typo in line 109, “unregulated”.
  3. Can the author provide the full list of APA genes in ACs and which of them are actually overlapped with transposable elements?
  4. Genome browser track example showing the enrichment of 3′ UTR isoforms of genes affected by alternative polyadenylation in osteoarthritic chondrocytes.
  5. Can the author further integrate their data with previous mRNA-seq (ArrayExpress E-MTAB-4304), that studied human knee OA cartilage? The difference is that the healthy control was from the relatively intact part of the OA patient. Such a model has been widely used (such as PMID: 30341348, PMID: 27686527) due to the limitation. Do they have any defects or biases in different assays? It can be further discussed by the authors.
  6. Is 10X Genomics single-cell 3'mRNA-seq data set sufficient for measuring APA? I wonder if the single-cell resolution has a better advantage.

Round 2

Reviewer 1 Report

Thanks for revising the manuscript. It has addressed all my concerns. I do not have any further questions. 

Author Response

We thank the reviewer for taking their time to read and comment on our manuscript.

Reviewer 2 Report

I thank the authors for their corrections and additions.

Author Response

We would like to thanks the reviewer for their comments on the manuscript.

Reviewer 3 Report

We appreciate the author's comprehensive response to the reviewer's questions. However, please address the following minor issues before the article can be accepted.

The authors compared the previous RNA-seq data (E-MTAB-4304) and found that one-third of the DEGs are consistent (R^2 0.8379), indicating the overall consistency and reliability of the data. However, the current study has also revealed new discoveries. In the new version of Fig1C, we can only see the frame of the scatter plot, but we cannot see the dots. The image may also be invisible due to formatting conversion. Please ensure that the content is complete.

It is necessary to emphasize again here: https://www.ebi.ac.uk/biostudies/arrayexpress/studies/E-MTAB-4304. The previous studies compared intact lateral condyle and damaged medial condyle. Therefore, aside from differences in consistency, these variations may also be caused by differences in batch, race, or the model. In addition to the transcriptome (RNA-seq), there are also OA studies on epigenome, including chromatin accessibility and DNA methylation (https://doi.org/10.1038/s41598-018-33779-zhttps://doi.org/10.1038/srep34460), relying on paired osteochondral samples. Conducting research on human clinical OA samples, as you know, obtaining healthy controls can be challenging. The author need to discuss the pros and cons of analyzing the omics characteristics of OA from paired osteochondral cartilage samples. This could greatly aid in the future direction of osteoarthritis clinical research. Investigating the integration of transcriptome and epigenome on the same batch of samples would make the study more meaningful if possible.

Author Response

We apologise for the formatting issue with figure 1 C. We have hopefully resolved this now.

We have added additional text in the discussion, commenting on the use of intact and fibrillated osteoarthritic tissue (line 356-374), in line with the reviewer's comment.